# Is there 'trustworthy' evidence for using manual therapy to treat patients with shoulder dysfunction?: A systematic review

Daniel W. Flowers[1], Brian T. Swanson[2], Stephen M. Shaffer[2], Derek J. Clewley[3,4]*, Sean P. Riley[5]

1 Doctor of Physical Therapy Program, Louisiana State University Health Sciences Center at Shreveport, Shreveport, Louisiana, United States of America, 2 Department of Rehabilitation Sciences, University of Hartford, West Hartford, Connecticut, United States of America, 3 Doctor of Physical Therapy Division, School of Medicine, Duke University, Durham, North Carolina, United States of America, 4 Duke Center for Excellence in Manual and Manipulative Therapy, Duke University, Durham, North Carolina, United States of America, 5 Hartford Healthcare Rehabilitation Network, Glastonbury, Connecticut, United States of America

* derek.clewley@duke.edu

**Editor:** Žiga Kozinc, Faculty of Health Sciences, University of Primorska, SLOVENIA

**Data Availability Statement:** All relevant data are within the manuscript and its Supporting Information files.

## Abstract

The primary objective of this review was to create a 'trustworthy,' living systematic review and meta-analysis for the application of manual therapy interventions in treating patients with shoulder dysfunction. Included studies were English-language randomized controlled trials published between 1/1/2010 and 8/3/2023, with searches performed in: PubMed, Cochrane Central Register of Controlled Trials (CENTRAL), CINHAL, ProQuest Nursing & Allied Health, EBSCO Medline, and PEDro. The population of focus included adults 18 years and older with musculoskeletal impairments related to shoulder dysfunction. Our primary outcomes included pain and region-specific outcome measures. We excluded trials, including participants having shoulder dysfunction resulting from surgery, radicular pain, instability/dislocation, fracture, lymphedema, and radiation. Our screening methodology was based upon a previously published 'trustworthy' systematic review protocol. This included the application of our PICOTS criteria in addition to screening for prospective clinical trial registration and following of prospective intent, as well as assessment of PEDro scores, risk-of-bias ratings, GRADE scoring, and examination of confidence in estimated effects. Twenty-six randomized controlled trials met our PICOTS criteria; however, only 15 of these were registered. Only three were registered prospectively. Two of these did not have discussions and conclusions that aligned with their primary outcome. The remaining single study was found to have a high risk-of-bias, meaning the remainder of the protocol could not be employed and that no randomized controlled trials could undergo further assessment or meta-analysis. The results of this systematic review indicate there are no 'trustworthy' randomized controlled trials examining the effectiveness of manual therapy interventions for the treatment of patients with shoulder dysfunction, as defined by the prospectively established methodology. Therefore, these findings signal that creating a 'trustworthy,' living systematic review on this clinically relevant topic is not yet possible due to a lack of 'trustworthy' randomized controlled trials.

**Funding:** The author(s) received no specific funding for this work.

**Competing interests:** I have read the journal's policy and the authors of this manuscript have the following competing interests: Brian Swanson and Sean Riley were authors on one of the papers identified in this review. This does not alter our adherence to PLOS ONE policies on sharing data and materials.

## Introduction

Systematic reviews (SRs) examining manual therapy's effectiveness in treating shoulder pathology have been methodologically diverse. These include the usage of broad [1–5] versus more narrow [6–12] definitions of the term "manual therapy," manual therapy being included in umbrella reviews along with other conservative interventions [4, 6, 12–21], and the inclusion of only one shoulder-related diagnosis in some reviews [3, 5, 8, 11, 13–20, 22] while others include many or prefer not to differentiate at all based upon diagnosis [6, 10, 23–25]. Additionally, a lack of evidence from which to build high-quality SRs has been repeatedly noted [1, 2, 8, 9, 11–14, 22–24, 26]. A cursory review of the literature for SRs studying the effectiveness of manual therapy as a treatment for shoulder dysfunction reveals 29 SRs, with or without meta-analyses, published since 2003 [1–29], including eight (~ 28%) published in the last five years [4, 5, 19–21, 25, 28, 29]. This indicates a high interest amongst researchers in providing clinicians with summary recommendations to implement in their clinical practice. The pace at which these SRs have been published when paired with the variation in the little evidence, leaves clinicians scrambling to keep current on whether the evidence supports manual therapy interventions they provide for patients with shoulder pathology. Our goal with this living SR was to provide clinicians with regularly updated recommendations based solely on those randomized controlled trials (RCTs) identified as 'trustworthy.'

Clinicians need easy access to accurate, clinically relevant preprocessed literature to guide their clinical decision-making. Journals subscribing to the clinical trials registration requirements put forth by the International Committee of Medical Journal Editors (ICMJE) do not necessarily follow their policies, as evidenced by only 33.7% of articles being prospectively registered in such journals (2016 data) [30]. Nevertheless, there was still evidence of RCTs being published despite not meeting these prospective registration requirements through 2020 [31]. This has led authors to conclude that "while many journals say they require prospective registration, they do not mean it" [32]. When paired with inadequate reporting of clinical trials methodology leading to increased risk-of-bias, clinicians are put in a predicament when assessing the relevance of the conclusions [33]. A lack of prospective registration in physical therapy RCTs limits the ability to determine the true rate of post-randomization bias existing therein [31], and could impact the reliability of the SRs that rely on the included RCTs [34]. Therefore, a protocol to examine the 'trustworthiness' of RCTs included in living SRs [35], which can be regularly updated, has been previously employed [36, 37] to ensure SRs disseminated to clinicians provide only the most trusted recommendations.

More recently, a Viewpoint by Riley et al. [38] emphasized the role such analyses play in the 'trustworthiness' of clinical evidence as it is implemented into clinical practice. Therefore, it remains to be seen whether the "strong" recommendations of previous authors [4] hold water when the findings of included RCTs are examined using the roadmap provided by Littlewood et al. [39] and Riley et al. [38]. Living SRs that employ protocols [35] aimed at establishing 'trustworthiness' are critical in helping clinicians implement sound evidence into their clinical practice [40].

Therefore, the aim of this review was to create a 'trustworthy,' living systematic review and meta-analysis that can provide clinicians with minimally biased, current recommendations on the state of the evidence for the treatment of shoulder dysfunction with manual therapy, and, if sufficient evidence is available, to provide findings related to specific shoulder pathologies (e.g., shoulder impingement, adhesive capsulitis, etc.) as in previous SRs [1, 7] since others have noted an umbrella term (e.g., "shoulder pain") is a limitation [23].

## Materials and methods

### Protocol and registration

The protocol for this SR was reviewed by the Louisiana State University Health Sciences Center at Shreveport Institutional Review Board and was considered exempt from oversight (STUDY00002449). In addition, the protocol for this SR was prospectively registered through the International Prospective Register of Systematic Reviews (CRD42023446571) [41]. The protocol follows that previously published by Riley et al. [35] for 'trustworthy,' living SRs with prospective modifications to the protocol that is more specific to this body region.

### Design

This SR was reported in agreement with the PRISMA 2020 statement and flow diagram [42].

### Eligibility criteria

This SR included English-language RCTs following PICOTS criteria [43]. The RCTs included *Patients* 18 years of age or older with musculoskeletal impairments consistent with an alteration in normal structure or function or an increase in pain or discomfort in the integument, muscles, bone, or joints of the body of an individual, which limits the function of the musculoskeletal system [44]. Joints of the spine referred to the cervical or thoracic regions, while peripheral joints referred to the shoulder (i.e., glenohumeral, scapulothoracic, and acromioclavicular joints)—manual therapy *Interventions* involved mobilization and manipulation to treat the spine or peripheral joints. Mobilization referred to a treatment that involved the clinician applying a sustained or oscillatory (at variable speeds and amplitudes) mechanical input to a joint to decrease pain and/or increase the range of motion [35]. Manipulation referred to a treatment involving the clinician applying a high-velocity, low-amplitude thrust to a joint to decrease pain and/or increase range of motion [35]. The manual therapy interventions were *Compared* to placebo, no treatment, other forms of conservative care, or in addition to other forms of conservative care. Other forms of conservative care included interventions like exercise and electrothermal modalities [35]. The primary *Outcomes* included pain (Visual Analog Scale [VAS] [45], Numeric Pain Rating Scale [NPRS]) and region-specific patient-reported outcome measures (PROMs) such as the Shoulder Pain and Disability Index (SPADI) [46, 47] and the Disabilities of the Arm, Shoulder and Hand (DASH) [48, 49] questionnaires. Potential secondary outcomes were measures of the patient's perceived improvement, such as the Global Rating of Change (GRoC) [50] or Single Assessment Numeric Evaluation (SANE) [51], and measures of positive (self-efficacy) [52] and negative (fear-avoidance [53], kinesiophobia [54]) psychological beliefs. Additionally, a modification to the published protocol was prospectively made and registered to add range of motion (ROM) as an outcome measure. *Time* of follow-up was [55]: Immediate = Closest to immediately following the intervention; Short-term = Closest to 1 month; Intermediate-term = closest to 6-months; and Long-term = closest to 12 months or longer. The types of *Studies* included RCTs. The following publication types were excluded: theses, dissertations, pilot/feasibility studies, published conference abstracts, cost-analysis studies, and secondary analyses of previously performed RCTs. Additionally, RCTs, including participants with shoulder dysfunction resulting from surgery, radicular pain, instability/dislocation, fracture, lymphedema, or radiation treatment, were excluded.

### Information sources

The following databases were searched: PubMed, Cochrane Central Register of Controlled Trials (CENTRAL), CINAHL, ProQuest Nursing & Allied Health, EBSCO Medline, and PEDro.

## Search strategy

The search parameters included RCTs from January 1, 2010, through August 3, 2023. The search was executed on August 3, 2023. The specific search strategy for this SR is available in the S1 Appendix: Search Strategy. A professional librarian assisted the authors in the development of the search strategy through the use of the Peer Review of Electronic Search Strategies (PRESS) checklist [56] according to each database used in the search [57, 58].

## Study records

**Data management.** Our Preferred Reporting Items for Systematic Reviews and Meta-Analyses (PRISMA) diagram is provided (Fig 1). Title screening was performed in EndNote (EndNote, Clarivate, Philadelphia, PA, USA). The results were imported into Covidence (Covidence systematic review software, Veritas Health Innovation, Melbourne, Australia; www.covidence.org) for abstract and full-text screening by two blinded authors (DWF and SMS).

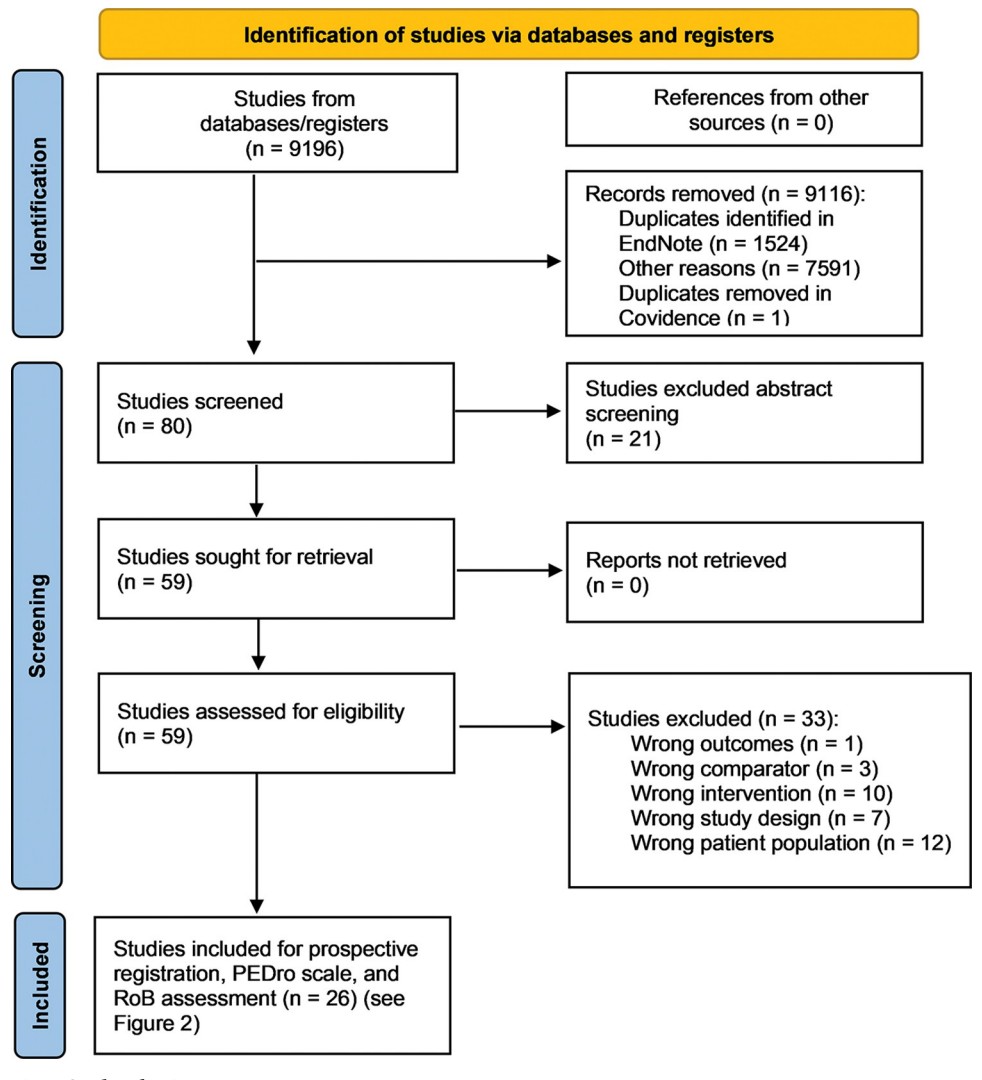

**Fig 1. Study selection.**

**Selection process.** We used the published protocol by Riley et al. [35] for the study selection process. Determination of prospective registration and whether the discussion and conclusion of the manuscript matched the registered protocol was accomplished by two blinded authors (DWF and SMS). We followed the same protocol to establish external validity by using the PEDro, ensuring the RCTs met the first criterion. Included studies also needed to have a PEDro score of at least six. The included studies had to have a moderate to low risk-of-bias as determined by the revised Cochrane risk-of-bias tool for randomized trials (RoB 2) [59]. Any remaining studies were required to have a moderate to high rating on the Grading of Recommendations, Assessment, Development, and Evaluations (GRADE) using Thoomes' methodology [60].

**Data collection process.** Our data collection process was also adopted from Riley et al. [35], specifically about recording registration status, whether the protocol was prospectively registered, determination of whether the findings of the study were consistent with the primary aim and outcome, determination of the PEDro Score, and risk-of-bias assessment (performed by two blinded reviewers (DWF and SMS).

**Data items.** The data items included our previously mentioned primary outcomes (i.e., pain and PROMs) and our potential secondary outcomes (e.g., ROM, GRoC, SANE, and positive and negative psychological beliefs). As previously defined, these were assessed across time and recorded as immediate, short-term, intermediate, and long-term [55]. Our plan was for data extraction to be completed in Covidence by two blinded reviewers (DWF and SMS) and analysis to be performed in RevMan 5.

**Data syntheses.** As indicated in our prospectively registered protocol and the published protocol by Riley et al. [35], data synthesis via a meta-analysis was planned.

**Confidence in cumulative evidence.** Per our published protocol [35], confidence in estimated effects was to be established through our inclusion of prospectively registered RCTs and reported findings consistent with the original registration and through the assessment of their external and internal validity (PEDro assessment) and RoB screening. Finally, the strength of the recommendation was to be rated using the GRADE. We made one modification to the previous protocol by Riley et al. [35] in case there were not enough homogeneous RCTs to synthesize using the GRADE evidence to recommendation framework. In this scenario, each study would be evaluated to assess the confidence in the estimated effects by examining $p$-values (statistical significance), estimated effects (differences larger than the minimally detectable change [MDC], minimal clinically important difference [MCID], and/or at least a moderate effect size), and precision (the size of the reported confidence interval and if the confidence intervals overlapped). This process has been previously described in the literature [38].

## Results

### Study selection (flow of studies)

The study selection process is outlined (Fig 1). Of the 9,196 studies identified and screened in EndNote and Covidence, only 80 were included for abstract screening in Covidence. Twenty-one studies were excluded, with 59 remaining for full-text review. Thirty-three studies were excluded via full-text review, with 26 studies moving forward to assessment of 'trustworthiness' via prospective registration, PEDro scale, and risk-of-bias assessment (Fig 2).

Fifteen studies were registered; however, only three were registered prospectively [61–63]. Kim et al. [61] was the only prospectively registered study where the discussion and conclusion matched the primary outcome. Therefore, was the only study to undergo PEDro and risk-of-bias assessment. Kheradmandi et al. [62] were excluded since the reviewers could not determine whether the primary outcome was used for the *a priori* power analysis. Three primary

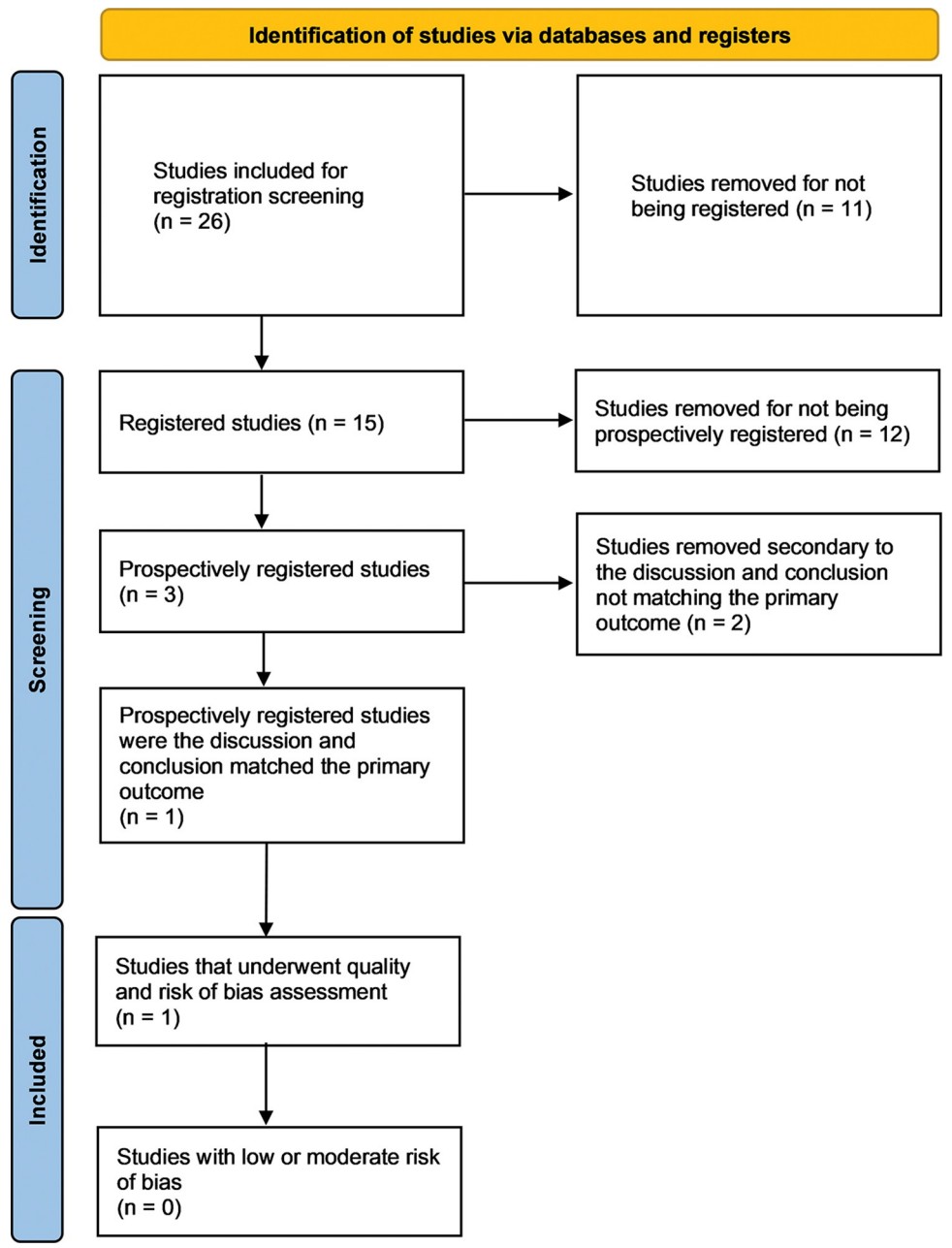

**Fig 2. Prospective registration, quality, and risk-of-bias screening.**

outcomes were registered, but the published article does not use the term primary or secondary when describing the outcomes, and the power analysis does not describe which variable was used for the power analysis. Therefore, determining which of the three variables was used in power analysis and should have been the focus of the discussion was not possible. Naranjo-Cinto et al. [63] were excluded because the reviewers could not determine which outcome was primary and whether it was used for the *a priori* power analysis. The published article listed the VAS as the primary outcome, but the registration listed the SPADI as the primary outcome. This prevented the reviewers from determining whether the discussion and conclusion matched the primary outcome.

**Table 1. Studies included for quality and risk of bias assessment.**

| Authors | Year | Title | Journal | Volume | Issue | Pages |
|---|---|---|---|---|---|---|
| Kim et al. [61] | 2020 | The application of the Neurac technique vs. manual therapy in patients during the acute phase of subacromial impingement syndrome: A randomized single-blinded controlled trial | J Back Musculoskelet Rehabil | 33 | 4 | 645–653 |

## Study characteristics

The study by Kim et al. [61] underwent quality and risk-of-bias assessment. The study characteristics are presented in Table 1. The studies excluded from this secondary screening process and the reasons for their exclusion are reported in Table 2.

## Risk of bias in studies

The PEDro scoring and RoB 2 ratings for the study by Kim et al. [61] are provided in Table 3. Domains 1 through 3 were rated as low risk-of-bias, while Domain 5: Selection of the Reported Result was scored as "some concerns." This rating resulted from the reviewers having no information regarding whether the data analysis plan, which was not part of the prospective clinical trials registration, originally included the statistical procedures presented in the published article (i.e., a 2 x 2 repeated measures ANOVA). Although, this statistical analysis is consistent with the design of the trial. This study was rated as high risk-of-bias on Domain 4: Measurement of the Outcome of the RoB 2 primarily due to the outcomes being assessed by the same investigators who provided the intervention. Although some outcomes would probably be protected from bias given the assessment methods (e.g., dynamometry for strength), others would be more susceptible to biased assessment (e.g., goniometry for range of motion). This resulted in an overall rating of high risk-of-bias. This precluded the study from being included in further analysis per our protocol.

## Data synthesis and confidence in cumulative evidence

The removal of the single remaining RCT due to high risk-of-bias resulted in us having no studies remaining for reporting outcomes, synthesis, reporting biases, or certainty of evidence reporting.

## Discussion

This review aimed to create a 'trustworthy,' living systematic review and meta-analysis that could provide clinicians with minimally biased, current recommendations on the state of the evidence for the manual therapy treatment of shoulder dysfunction. Unfortunately, we could not identify any RCTs investigating the effect of manual therapy on shoulder dysfunction that met our PICOTS question and were prospectively registered, whose discussion and conclusions matched the primary outcome and passed quality and risk-of-bias screening. Only one RCT by Kim et al. [61] could proceed to risk-of-bias assessment but was excluded from the review due to a high risk-of-bias rating. These results are unfortunate, given that shoulder pain was recently found to have an annual global incidence rate of 7.7 and 62 per 1,000 people (median 37.8) [87]. This strongly indicates that having access to high-quality, trustworthy evidence for managing shoulder dysfunction is imperative.

When we consider previous SRs examining the use of manual therapy for treating persons with shoulder dysfunction, it is evident that problems identified in our SR are common. A lack of prospective clinical trial registration and high bias levels are observed in the RCTs already included in these previous SRs. For example, Satpute et al. [25] included 25 RCTs (published

**Table 2. Studies excluded with reasons.**

| Authors | Year | Title | Journal | Volume | Issue | Pages | Reason |
|---|---|---|---|---|---|---|---|
| Cook et al. [64] | 2014 | The addition of cervical unilateral posterior-anterior mobilisation in the treatment of patients with shoulder impingement syndrome: A randomised clinical trial | Man Ther | 19 | 1 | 18–24 | Retrospective registry |
| Coronado et al. [65] | 2015 | The comparative effects of spinal and peripheral thrust manipulation and exercise on pain sensitivity and the relation to clinical outcome: A mechanistic trial using a shoulder pain model | J Orthop Sports Phys Ther | 45 | 4 | 252–264 | Unregistered |
| da Silva et al. [66] | 2019 | Immediate effects of spinal manipulation on shoulder motion range and pain in individuals with shoulder pain: A randomized trial | J Chiropr Med | 18 | 1 | 19–26 | Unregistered |
| Eliason et al. [67] | 2021 | Guided exercises with or without joint-mobilization or no treatment in patients with subacromial pain syndrome | J Rehabil Med | 53 | 5 | 2765 | Retrospective registry |
| Grimes et al. [68] | 2019 | The comparative effects of upper thoracic spine thrust manipulation techniques in individuals with subacromial pain syndrome: A randomized clinical trial | J Orthop Sports Phys Ther | 49 | 10 | 716–724 | Retrospective registry |
| Guimarães et al. [69] | 2016 | Immediate effects of mobilization with movement vs sham technique on range of motion, strength, and function in patients with shoulder impingement syndrome: Randomized clinical trial | J Manipulative Physiol Ther | 39 | 9 | 605–615 | Retrospective registry |
| Gutiérrez-Espinoza et al. [70] | 2023 | Effectiveness of scapular mobilization in people with subacromial impingement syndrome: A randomized controlled trial | Ann Phys Rehabil Med | 66 | 5 | 101744 | Retrospective registry |
| Haider et al. [71] | 2018 | Comparison of conservative exercise therapy with and without Maitland Thoracic Manipulative therapy in patients with subacromial pain: Clinical trial | J Pak Med Assoc | 68 | 3 | 381–387 | Unregistered |
| Haik et al. [72] | 2017 | Short-term effects of thoracic spine manipulation on shoulder impingement syndrome: A randomized controlled trial | Arch Phys Med Rehabil | 98 | 8 | 1594–1605 | Retrospective registry |
| Haik et al. [73] | 2014 | Scapular kinematics pre- and post-thoracic thrust manipulation in individuals with and without shoulder impingement symptoms: A randomized controlled study | J Orthop Sports Phys Ther | 44 | 7 | 475–487 | Unregistered |
| Kardouni et al. [74] | 2015 | Thoracic spine manipulation in individuals with subacromial impingement syndrome does not immediately alter thoracic spine kinematics, thoracic excursion, or scapular kinematics: A randomized controlled trial | J Orthop Sports Phys Ther | 45 | 7 | 527–538 | Unregistered |
| Kardouni et al. [75] | 2015 | Immediate changes in pressure pain sensitivity after thoracic spinal manipulative therapy in patients with subacromial impingement syndrome: A randomized controlled study | Man Ther | 20 | 4 | 540–546 | Unregistered |
| Khalil et al. [76] | 2022 | Comparison of Mulligan technique versus muscle energy technique in patients with adhesive capsulitis | J Pak Med Assoc | 72 | 2 | 211–215 | Retrospective registry |
| Kheradmandi et al. [62] | 2021 | Comparison between dry needling plus manual therapy with manual therapy alone on pain and function in overhead athletes with scapular dyskinesia: A randomized clinical trial | J Bodyw Mov Ther | 26 | n/a | 339–346 | Discussion and conclusion did not match primary outcome |
| Lluch et al. [77] | 2018 | Effects of an anteroposterior mobilization of the glenohumeral joint in overhead athletes with chronic shoulder pain: A randomized controlled trial | Musculoskelet Sci Pract | 38 | n/a | 91–98 | Retrospective registry |
| Menek et al. [78] | 2019 | The effect of Mulligan mobilization on pain and life quality of patients with Rotator cuff syndrome: A randomized controlled trial | J Back Musculoskelet Rehabil | 32 | 1 | 171–178 | Unregistered |
| Michener et al. [79] | 2015 | Validation of a sham comparator for thoracic spinal manipulation in patients with shoulder pain | Man Ther | 20 | 1 | 171–175 | Unregistered |
| Mintken et al. [80] | 2016 | Cervicothoracic manual therapy plus exercise therapy versus exercise therapy alone in the management of individuals with shoulder pain: A multicenter randomized controlled trial | J Orthop Sports Phys Ther | 46 | 8 | 617–628 | Retrospective registry |

*(Continued)*

**Table 2.** (Continued)

| Authors | Year | Title | Journal | Volume | Issue | Pages | Reason |
|---|---|---|---|---|---|---|---|
| Naranjo-Cinto et al. [63] | 2022 | Real versus sham manual therapy in addition to therapeutic exercise in the treatment of non-specific shoulder pain: A randomized controlled trial | J Clin Med | 11 | 15 | 4395 | Discussion and conclusion did not match primary outcome |
| Pekgoz et al. [81] | 2020 | Comparison of mobilization with supervised exercise for patients with subacromial impingement syndrome | Turk J Phys Med Rehabil | 66 | 2 | 184–192 | Unregistered |
| Riley et al. [82] | 2015 | Short-term effects of thoracic spinal manipulations and message conveyed by clinicians to patients with musculoskeletal shoulder symptoms: A randomized clinical trial | J Man Manip Ther | 23 | 1 | 3–11 | Retrospective registry |
| Satpute et al. [83] | 2015 | Efficacy of hand behind back mobilization with movement for acute shoulder pain and movement impairment: A randomized controlled trial | J Manip Physiol Ther | 38 | 5 | 324–334 | Retrospective registry |
| Suri et al. [84] | 2013 | Comparative Study on the effectiveness of Maitland Mobilization Technique Versus Muscle Energy Technique in Treatment of Shoulder Adhesive Capsulitis | Indian J Physiother Occup Ther | 7 | 4 | 44932 | Unregistered |
| Teys et al. [85] | 2013 | One-week time course of the effects of Mulligan's mobilisation with movement and taping in painful shoulders | Man Ther | 18 | 5 | 372–377 | Unregistered |
| Yiasemides et al. [86] | 2011 | Does passive mobilization of shoulder region joints provide additional benefit over advice and exercise alone for people who have shoulder pain and minimal movement restriction? A randomized controlled trial | Phys Ther | 91 | 2 | 178–189 | Retrospective registry |

through January 2021) in their SR, of which only six (24%) were registered. The prospective/retrospective status of these six existing registrations is unclear in the review by Satpute et al. [25]. The SR published by Minns Lowe et al. [19] included 30 RCTs, with only four rated as having low bias levels. To the authors' credit, these were the only studies included in the data synthesis. Desjardins et al. [2] included 21, of which approximately 76% had a high risk of bias, with approximately 81% not providing a registration number. Additionally, two-year delays in publication are commonly observed in several of the previous SRs [3, 19, 21]. The most significant delay was observed in the SR by Gebremariam et al. [16], where the search was conducted through March 2009, but the SR itself was not published until November 2013.

Including RCTs with unestablished 'trustworthiness' in SRs may require revision of the SRs if the RCTs are proven 'untrustworthy' (e.g., published inconsistent with the prospective research record) in the future [88]. One method of preventing this unseemly occurrence would be to avoid such RCTs from being included in SRs in the first place. We cannot assume that all studies in SRs are 'trustworthy,' and establishing methodologies to prove the 'trustworthiness' of RCTs included in SRs is warranted [89]. Our results indicate that 29 published SRs [1–29] made it through the entirety of the peer-review process despite containing shortcomings that prohibit a discerning reader from determining the quality of the outcomes and conclusions.

**Table 3. PEDro and RoB Assessment.**

| Authors | Official PEDro Scale | PEDro Criterion 1 | PEDro Scores | RoB 2 Randomization Process | RoB 2 Deviations from the intended interventions | RoB 2 Missing outcome data | RoB 2 Measurement of the outcome | RoB 2 Selection of the reported result | RoB 2 Overall Risk |
|---|---|---|---|---|---|---|---|---|---|
| Kim et al. 2020 [61] | Yes | Yes | 6 | Low | Low | Low | High | Some Concerns | High |

## How much high-quality data supports manual therapy use for this population?

In 2020, Pieters et al. [4] published a review of SRs surveying the evidence supporting interventions, including manual therapy, in treating persons with subacromial shoulder pain. The authors concluded there was "strong" evidence supporting the use of manual therapy when paired with exercises; however, their definition of manual therapy was broader than the one included herein, including both neurodynamic and soft tissue mobilization techniques, making it difficult to determine precisely which interventions were helpful while simultaneously raising the question if these RCTs were too heterogeneous to be synthesized. In response to the review published by Pieters et al. [4], Littlewood et al. [39] published a Letter to the Editor questioning the review's conclusions, namely the lack of consideration given to confidence in estimated effects, including a lack of analysis of wide confidence intervals observed and clinically important differences, apart from *p*-values. Minns Lowe et al. [19] pointed out the importance of analyzing whether studies included in the SR had "clinically meaningful" findings.

A lack of 'trustworthy' evidence to guide the use of manual therapy by clinicians is not isolated to shoulder dysfunction. Using the same protocol [35] as our review, Riley et al. [36] were only able to identify a single 'trustworthy' RCT [90] guiding the use of manual therapy for treating patients with non-radicular cervical spine impairments. Riley et al. [37] encountered similar results when investigating when manual therapy affects quantitative sensory testing and patient-reported outcome measures with varying musculoskeletal impairments, with only three RCTs [90–92] able to be included in the 'trustworthy' review. These results indicate the problem is not isolated to studies investigating the use of manual therapy for treating patients with dysfunction of a single joint but may apply to trials investigating the treatment regardless of pathology/region. Even more concerning is the possibility this lack of 'trustworthy' guidance may apply to many interventions employed by clinicians. Most concerning is the invasion of 'untrustworthy' RCTs into a clinical practice guideline (CPG). For example, O'Connell et al. [89] reported the CPG for the treatment of patients with acute and chronic low back pain [93] included an 'untrustworthy' study by Monticone et al. [94]. O'Connell et al. [89] found six areas of concern out of 11 criteria considered, including prospective registration and plausibility of the findings, when examining the study by Monticone et al. [94]. Additionally, the data therein closely resembled that of two other studies by Monticone, both of which have been retracted and are cited herein to emphasize the impact 'untrustworthy' data can have on professional treatment recommendations [95–98]. For example, the study [94] was included in the CPG [93], representing 25% of the papers used to determine the evidence supporting the inclusion of general exercise for treating low back pain. Despite being the only paper to show beneficial effects, the CPG recommends prescribing available exercises. Manual therapy may very well be an effective and efficient intervention method; we do not have strong evidence supporting formal recommendations of the treatment [99] due to a lack of confidence in previous findings and objective data.

## Is our protocol too rigorous?

The International Society of Physiotherapy Journal Editors has embraced the ICMJE requirement for prospective clinical trial registration since 2012 [100]. In 2013, Pinto et al. [101] reported that only 34% of clinical trials of physical therapy interventions were registered, and only 6% prospectively. Forty-seven percent were found to have selective outcome reporting. In 2023, Silva et al. [102] published data from 2019 indicating that 63% of trials were registered; however, only 18% were done so prospectively. Although registration and prospective registration rates have improved, the most concerning finding by Silva et al. [102] was that the rate of

selective outcome reporting, a form of post-randomization bias, had *increased* to 73%. This problem is not isolated to physical therapy literature. A recent article published in *Nature* discusses the alarming rate of research falsification across medical research when looking at the publically available research data [88], a level of rigor that has not been employed in this SR's 'trustworthy' process. Carlisle [103] in 2020 identified that 44% of the raw data examined was faked or fatally flawed, calling these 'zombie' trials because they looked like real research but were empty vessels impersonating research. Given that physical therapy clinical practice has not meaningfully changed in the past 30 years [104], increased rigor involving moderate to high-quality evidence that can be confidently translated into accurate strong clinical practice recommendations is needed.

### Is our protocol not rigorous enough?

Besides generating 'untrustworthy' findings, RCTs that fail to register and adhere to their initial research intent prospectively can become more misleading as they are incorporated into systematic reviews, which have an even greater capacity to impact clinical practice. Richard van Noorden has quoted Žarko Alfirević as stating, "'*An untrustworthy systematic review is far more dangerous than an untrustworthy primary study* [88].'" Establishing 'trustworthy' RCTs supporting the application of manual therapy interventions for specific impairments is therefore critical as the profession aims to improve evidence-based practice and advocate for our services at the national and local levels. Without 'trustworthy' data to back up our care plans, we will not have adequate means of demonstrating our value to society and, most importantly, our patients.

### Limitations

The primary limitation of this SR is the absence of any RCTs that could be included for practice recommendations to be made. We could not implement our full protocol, including GRADE assessment and an interpretation in confidence in estimated effects reported by RCTs, given that the last RCT was eliminated at the risk-of-bias stage of our screening process. Physical therapists' use of recommended and non-recommended treatments in their clinical practice has not changed in the past 30 years [104]. One possible reason for this is that the quality of RCTs and the SRs that synthesize them have generally been critically low to low [105]. It has been consistently recommended in SRs that research quality needs to improve to answer the clinically relevant questions of practicing clinicians. This continued lack of 'trustworthy' evidence prevents this SR from being able to provide answers to this particular question of clinical importance.

### Conclusions

The goal of this SR was to establish 'trustworthy' recommendations for applying manual therapy in treating patients with shoulder dysfunction. Our methodology could not identify any RCTs meeting our inclusion/exclusion criteria that passed the screening process aimed at determining 'trustworthiness,' leading to a complete absence of RCTs from which to derive clinical recommendations. With manual therapy being one of the most prescribed interventions in physical therapy practice and shoulder dysfunction being a common impairment in our patients, creating strong, 'trustworthy' RCTs investigating the effectiveness of manual therapy in treating patients with shoulder dysfunction is of utmost importance.

## Supporting information

**S1 Appendix. Search strategy.**
(PDF)

**S2 Appendix. PRISMA checklist.**
(PDF)

## Acknowledgments

The authors would like to thank Nicholas Wharton, MLIS from the University of Hartford, for his assistance in developing the search strategy.

## Author Contributions

**Conceptualization:** Daniel W. Flowers, Brian T. Swanson, Stephen M. Shaffer, Sean P. Riley.

**Data curation:** Daniel W. Flowers, Brian T. Swanson, Stephen M. Shaffer, Sean P. Riley.

**Formal analysis:** Daniel W. Flowers, Brian T. Swanson, Stephen M. Shaffer, Sean P. Riley.

**Investigation:** Daniel W. Flowers, Brian T. Swanson, Stephen M. Shaffer, Sean P. Riley.

**Methodology:** Daniel W. Flowers, Brian T. Swanson, Sean P. Riley.

**Project administration:** Daniel W. Flowers, Sean P. Riley.

**Resources:** Daniel W. Flowers, Derek J. Clewley, Sean P. Riley.

**Software:** Daniel W. Flowers, Sean P. Riley.

**Supervision:** Daniel W. Flowers, Sean P. Riley.

**Validation:** Daniel W. Flowers, Sean P. Riley.

**Visualization:** Sean P. Riley.

**Writing – original draft:** Daniel W. Flowers, Brian T. Swanson, Stephen M. Shaffer, Sean P. Riley.

**Writing – review & editing:** Daniel W. Flowers, Brian T. Swanson, Stephen M. Shaffer, Derek J. Clewley, Sean P. Riley.

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
