## [Decision Letter · Decision Letter 0]

28 Nov 2023

PONE-D-23-32393Is there ‘trustworthy’ evidence for using manual therapy to treat patients with shoulder dysfunction?PLOS ONE

Dear Dr. Clewley,

Thank you for submitting your manuscript to PLOS ONE. After careful consideration, we feel that it has merit but does not fully meet PLOS ONE’s publication criteria as it currently stands. Therefore, we invite you to submit a revised version of the manuscript that addresses the points raised during the review process.

We look forward to receiving your revised manuscript.

Kind regards,

Žiga Kozinc

Academic Editor

PLOS ONE

Journal Requirements:

2. Please identify your study as "systematic review" in the title of your manuscript.

"I have read the journal's policy and the authors of this manuscript have the following competing interests: Brian Swanson and Sean Riley were authors on one of the papers identified in this review. "

Reviewers' comments:

Reviewer's Responses to Questions

**Comments to the Author**

1. Is the manuscript technically sound, and do the data support the conclusions?

Reviewer #1: Yes

Reviewer #2: Yes

Reviewer #3: Partly

2. Has the statistical analysis been performed appropriately and rigorously? 

Reviewer #1: Yes

Reviewer #2: N/A

Reviewer #3: No

3. Have the authors made all data underlying the findings in their manuscript fully available?

Reviewer #1: Yes

Reviewer #2: Yes

Reviewer #3: No

4. Is the manuscript presented in an intelligible fashion and written in standard English?

Reviewer #1: Yes

Reviewer #2: Yes

Reviewer #3: Yes

5. Review Comments to the Author

Reviewer #1: Congratulations on your study. I think that this is a very interesting study and should be accepted for publication. The manuscript is technically sound, the methodology used is appropriate and the manuscript presented in an intelligible fashion and written in standard English.

Reviewer #2: Thank you for the opportunity to review this manuscript, “Is there ‘trustworthy’ evidence for using manual therapy to treat patients with shoulder dysfunction?” The authors present a systematic review of manual therapy for the treatment and management of patients with shoulder dysfunction. Multiple efforts to promote rigor were employed to include a prior protocol registration, multiple databases were searched, and clear and a concise PICOTS format. The methods resulted in no RCTs meeting full criteria and thus the authors were unable to execute the protocol in full. The conclusions are appropriate to the findings and do not overstep the results.

I thank the authors for making this project to completion, as some may deem this a negative result, this is highly relevant to the management of shoulder dysfunction and more broadly musculoskeletal condition management. The is a well executed systematic review with multiple layers of inclusion addressing risk of bias and quality that unfortunately reflects the nature of RCT quality for shoulder dysfunction using manual/manipulative therapy treatments. The search strategies by database are presented transparently.

General comments:

I thank the authors to for considering the PRISMA checklist for guidance on reporting. Please revise the title, to include ‘systematic review’ in the title.

Line 88-90. The sentences describing “Spinal…” and “The peripheral joint…” appear incomplete in what they are describing or perhaps worded awkwardly. Are you perhaps highlighting that both spinal and peripheral joint etiologies are included Please consider revising for clarity.

Line 91-93 Please provide citing source(s) for the definition of mobilization.

Line 93-95 Please provide citing source(s) for the definition of manipulation. Further a therapist, chiropractor, or osteopath may all be trained in manipulation procedures.

Line 96-97. What are examples of “other forms of conservative care”?

Line 102-103 Please provide supporting citations for “…measures of positive (self-efficacy) and negative (fear-avoidance, kinesiophobia) psychological beliefs.” in similar fashion to SPADI and DASH.

Line 150-159 is seemingly a duplication of content described line 97-108 rather than a concise description of extractable elements of interest. Please consider revising for conciseness.

Line 162-166 Results are bleeding into the planned methods. Report planned methods here and report results or inability to derive results in an appropriate sub-section of the Results. Methods are well-described in the published protocol and so a succinct description is appropriate.

Line 168-177 Similar to prior comment, Report the planned methods here and report results below. This section should correspond to a reporting in the Results section.

Fig1/Fig2/Table 1. After reading the methods, I expected to find included studies (Table 1) to report specific shoulder dysfunction conditions included in the RCT, manual/manipulative therapies used, the intervention description and the comparator group intervention, time frame for outcome, specific outcomes used and summary conclusion yet, what is presented is merely a citation. Was this not included because extraction of these data elements would have been completed with COVIDENCE?

On review of the methods, I understand that Kim et al. did in fact not make it to final inclusion due to finding a rating of high risk-of-bias. Several items – 1) fig 2 gives the impression, Kim et al meets inclusion – another level is required to reflect that Kim is moved to exclusion for the final stage of the protocol. 2) Line 232-234 should be a new section of results matching proposed methods: Data Syntheses and Confidence of Cumulative Evidence. Please revise for clarity.

Table 1 and Table 2. Reporting columns for volume, issue, and pages seems redundant since the article citation is also demarcated and can be found in the bibliography. Recommend revising and simplifying Table 2. If more studies met inclusion and prospective registration, knowing the journal could be a telling feature for pattern recognition.

Table 3. Official PEDro score value would be more valuable than a yes/no criteria. By virtue of the study making it inclusion and presented in Table 3, PEDro score reporting is ‘yes’.

I find the PRISMA flow charts digestible and logically with the unique methodological approach of requiring to levels of inclusion criteria. I defer to the editorial staff in determining if any formatting changes are required for the PRIMSA charts –

Fig 1 and Fig 2 have the same left side labels, which could confusing to some readers. As noted above, my interpretation is that an additional inclusion drop box should be used to notate the final exclusion of Kim et al.

I find the use of questions in the discussion unusual (line 271-272, line 309-310). Perhaps consider these as subheadings for a section if you are unable to revise in another manner?

Do the authors view any limitations of their methodological approach? Limitations at this time are limited to the absence of RCTs that met full inclusion criteria and thus an inability to execute the full protocol.

Reviewer #3: Dear authors:

First of all I must congratulate you on the great research work carried out. I think it is a great topic and a great systematic review with all the requirements for a review. I would like to suggest some areas for improvement:

- You should include a flow chart so that we readers can see how you have selected the articles and how many losses you have obtained in this search. I strongly suggest that you insert a specific flowchart for reviews and meta-analyses.

- You talk about conducting a systematic review and meta-analysis study, but you have not conducted the meta-analysis, so in the result section you should include all the data derived from the meta-analysis, as well as the diagrams, graphs and/or figures derived from it. So far, the article would only be a systematic review, so you have two options: leave it as it is and call it a review or add the entire meta-analysis.

We look forward to your decision.

Regards

6. PLOS authors have the option to publish the peer review history of their article (what does this mean?). If published, this will include your full peer review and any attached files.

Reviewer #1: **Yes: **Paraskevopoulos Eleftherios

Reviewer #2: No

Reviewer #3: **Yes: **Jorge Velázquez Saornil

---

## [Author Response · Author response to Decision Letter 0]

19 Dec 2023

Dear Editors, 

We would like to re-submit for your consideration for publication in PLOS One our revised manuscript entitled, “Is there ‘trustworthy’ evidence for using manual therapy to treat patients with shoulder dysfunction?: A systematic review.” We thank you for the opportunity to revise our manuscript and found your input helpful in making this manuscript a better product. We have attempted to respond to all comments from the editor and reviewers as outlined in detail below. 

In the request for revisions, the editors asked we update our competing interests statement. Our revised statement is below: 

I have read the journal's policy and the authors of this manuscript have the following competing interests: Brian Swanson and Sean Riley were authors on one of the papers identified in this review. This does not alter our adherence to PLOS ONE policies on sharing data and materials.

Sincerely, 

Derek Clewley, PT, DPT, PhD

Duke University

Assistant Professor, Doctor of Physical Therapy Division

Derek.clewley@duke.edu

Comments from Editor and Reviewers with Responses from the Authors: 

Response to the Journal: We have referenced the above links to make edits to the manuscript. This includes adding line numbers to the title and authors’ page. Additionally, we have changed the heading for “Methods” to “Materials and methods,” the way we refer to our figures in the text and emboldening our table titles and adding spacing before each table. We have made separate heading for Results, Discussion, and Conclusions per the submission guidelines. 

2. Please identify your study as "systematic review" in the title of your manuscript.

Response to the Journal: We have edited the title of the manuscript to reflect that it is a systematic review.

"I have read the journal's policy and the authors of this manuscript have the following competing interests: Brian Swanson and Sean Riley were authors on one of the papers identified in this review. "

Response to the Journal: We have included this revised statement in our cover letter for the revised version of the manuscript. 

Response to the Journal: We have included our reason for citing the retracted articles, along with their respective retraction notices, in the manuscript text. We have also made changes to the reference list, ensuring we include the retraction notice and status of each article referenced. 

5. Review Comments to the Author

Reviewer #1: Congratulations on your study. I think that this is a very interesting study and should be accepted for publication. The manuscript is technically sound, the methodology used is appropriate and the manuscript presented in an intelligible fashion and written in standard English.

Response to the Reviewer: We thank the reviewer for their kind comments and effort in reviewing this manuscript. We appreciate the feedback and comments that are provided. 

Reviewer #2: Thank you for the opportunity to review this manuscript, “Is there ‘trustworthy’ evidence for using manual therapy to treat patients with shoulder dysfunction?” The authors present a systematic review of manual therapy for the treatment and management of patients with shoulder dysfunction. Multiple efforts to promote rigor were employed to include a prior protocol registration, multiple databases were searched, and clear and a concise PICOTS format. The methods resulted in no RCTs meeting full criteria and thus the authors were unable to execute the protocol in full. The conclusions are appropriate to the findings and do not overstep the results.

Response to the Reviewer: Thank you for the comments specifically related to the rigor of our methodology and presentation of the findings. We hope to use the comments you provided in your review to improve the overall clarity of our conclusions. 

I thank the authors for making this project to completion, as some may deem this a negative result, this is highly relevant to the management of shoulder dysfunction and more broadly musculoskeletal condition management. The is a well executed systematic review with multiple layers of inclusion addressing risk of bias and quality that unfortunately reflects the nature of RCT quality for shoulder dysfunction using manual/manipulative therapy treatments. The search strategies by database are presented transparently.

Response to the Reviewer: Thank you for your comments regarding the transparency of our research process. Although we were disappointed with the results of our review, we hope readers are still able to discern the need to improve the overall quality of evidence available to clinicians as they provide care to their patients. In our Limitations, we now discussed the issue of stalled progress of the inclusion of best evidence into clinical practice as it relates to low levels evidence in the field. 

General comments:

I thank the authors to for considering the PRISMA checklist for guidance on reporting. Please revise the title, to include ‘systematic review’ in the title.

Response to the Reviewer: We have edited the title to reflect that the manuscript is in fact a systematic review. 

Line 88-90. The sentences describing “Spinal…” and “The peripheral joint…” appear incomplete in what they are describing or perhaps worded awkwardly. Are you perhaps highlighting that both spinal and peripheral joint etiologies are included Please consider revising for clarity.

Response to the Reviewer: We have edited the phrasing of these lines to improve clarity. We examined both spinal (cervical and thoracic) and shoulder (i.e., glenohumeral, scapulothoracic, and acromioclavicular joints) manual therapy treatments for managing patients with shoulder dysfunction.

Line 91-93 Please provide citing source(s) for the definition of mobilization.

Response to the Reviewer: We have cited our previously published protocol here since this is where this definition was initially introduced. This definition was a combination of both Maitland and Kaltenborn to be inclusive of different philosophies of manual therapy treatment.

Line 93-95 Please provide citing source(s) for the definition of manipulation. Further a therapist, chiropractor, or osteopath may all be trained in manipulation procedures.

Response to the Reviewer: We have cited our previously published protocol here since this is where this definition was initially introduced. This definition was a combination of both Maitland and Kaltenborn to be inclusive of different philosophies of manual therapy treatment. We have also replaced “therapist” with “clinician” to include other practitioners that use these techniques.

Line 96-97. What are examples of “other forms of conservative care”?

Response to the Reviewer: From our previous protocol, we state, “The manual therapy interventions as described in this protocol will be Compared to placebo, no treatment, or other forms of conservative care such as manual therapy with exercise, therapeutic exercise, and electrothermal modalities.” We have included this description and cited our published protocol.

Line 102-103 Please provide supporting citations for “…measures of positive (self-efficacy) and negative (fear-avoidance, kinesiophobia) psychological beliefs.” in similar fashion to SPADI and DASH.

Response to the Reviewer: We have included the following references for these outcome measures as requested. This has led to changes in our reference list and reference numbers for the manuscript as a result. 

Schwarzer, R., & Jerusalem, M. (1995). Generalized Self-Efficacy scale. In J. Weinman, S. Wright, & M. Johnston, Measures in health psychology: A user’s portfolio. Causal and control beliefs (pp. 35-37). Windsor, UK: NFER-NELSON.

Waddell G, Newton M, Henderson I, Somerville D, Main CJ. A Fear-Avoidance Beliefs Questionnaire (FABQ) and the role of fear-avoidance beliefs in chronic low back pain and disability. Pain 1993; 52:157-168

Woby SR, Roach NK, Urmston M, Watson PJ. Psychometric properties of the TSK-11: a shortened version of the Tampa Scale for Kinesiophobia. Pain 2005;117:137-44.

Sean P Riley, Vincent Tafuto, Mark Cote, Jean-Michel Brismée, Alexis Wright & Chad Cook (2019) Reliability and relationship of the fear-avoidance beliefs questionnaire with the shoulder pain and disability index and numeric pain rating scale in patients with shoulder pain, Physiotherapy Theory and Practice, 35:5, 464-470, DOI: 10.1080/09593985.2018.1453004

Franchignoni, Franco, et al. Minimal Clinically Important Difference of the Disabilities of the Arm, Shoulder and Hand Outcome Measure (DASH) and Its Shortened Version (QuickDASH)J Orthop Sports Phys Ther 2014;44(1):30-39. Epub 30 October 2013. doi:10.2519/jospt.2014.4893

Line 150-159 is seemingly a duplication of content described line 97-108 rather than a concise description of extractable elements of interest. Please consider revising for conciseness.

Response to the Reviewer: This section (Data items) has been presented more concisely and is no longer identical to the eligibility criteria enumerated previously. 

Line 162-166 Results are bleeding into the planned methods. Report planned methods here and report results or inability to derive results in an appropriate sub-section of the Results. Methods are well-described in the published protocol and so a succinct description is appropriate.

Response to the Reviewer: We removed the line referencing our inability to complete our methodology, and simply report that which was planned. Our results now report this inability to follow the synthesis component of the protocol. 

Line 168-177 Similar to prior comment, Report the planned methods here and report results below. This section should correspond to a reporting in the Results section.

Response to the Reviewer: Similar to our response above, we have reported our original intent per the previously published methodology and have signified our inability to complete the methodology in our results. This illustrates that research quality is not improving despite consistent recommendations that this is necessary, which may be related to the lack of improvement in clinical outcomes that has occurred in the absence of these improvements.

Fig1/Fig2/Table 1. After reading the methods, I expected to find included studies (Table 1) to report specific shoulder dysfunction conditions included in the RCT, manual/manipulative therapies used, the intervention description and the comparator group intervention, time frame for outcome, specific outcomes used and summary conclusion yet, what is presented is merely a citation. Was this not included because extraction of these data elements would have been completed with COVIDENCE?

Response to the Reviewer: Table 1 is “Studies Included for Quality and Risk of Bias Assessment.” This study made it through the prospective validity check and was eliminated secondary to the risk of bias; therefore, data elements were not extracted. 

On review of the methods, I understand that Kim et al. did in fact not make it to final inclusion due to finding a rating of high risk-of-bias. Several items – 1) fig 2 gives the impression, Kim et al meets inclusion – another level is required to reflect that Kim is moved to exclusion for the final stage of the protocol. 2) Line 232-234 should be a new section of results matching proposed methods: Data Syntheses and Confidence of Cumulative Evidence. Please revise for clarity.

Response to the Reviewer: We have edited Figure 2 to indicate that no studies had a low or moderate risk of bias to be included in the SR for data synthesis, etc. We have also created a new section of the manuscript to reflect that this aspect of the SR and planned meta-analysis could not be completed because of the removal of Kim et al. due to high risk of bias. 

Table 1 and Table 2. Reporting columns for volume, issue, and pages seems redundant since the article citation is also demarcated and can be found in the bibliography. Recommend revising and simplifying

Response to the Reviewer: The presentations of the tables follow the way the RCTs have been presented in our previous publications using the same protocol for the sake of consistency. This also prevents the reader from having to search the reference list as they read through each stage of the screening process. We are more than willing to make specific changes that the journal suggests. 

 Table 2. If more studies met inclusion and prospective registration, knowing the journal could be a telling feature for pattern recognition.

Response to the Reviewer: This is a very interesting point. Although this was not our research question, this work has previously been published.

The Unknown Prevalence of Postrandomization Bias in 15 Physical Therapy Journals: A Methods Review J Orthop Sports Phys Ther 2021;51(11):542-550. Epub 21 Sep 2021. doi:10.2519/jospt.2021.10491 

Table 3. Official PEDro score value would be more valuable than a yes/no criteria. By virtue of the study making it inclusion and presented in Table 3, PEDro score reporting is ‘yes’.

Response to the Reviewer: The official PEDro score is reported in Table 3 and is “6” (fourth column from the left). 

I find the PRISMA flow charts digestible and logically with the unique methodological approach of requiring to levels of inclusion criteria. I defer to the editorial staff in determining if any formatting changes are required for the PRIMSA charts –

Response to the Reviewer: Thank you for finding our PRISMA flowchart helpful despite the uniqueness of our methodology. 

Fig 1 and Fig 2 have the same left side labels, which could confusing to some readers. As noted above, my interpretation is that an additional inclusion drop box should be used to notate the final exclusion of Kim et al.

Response to the Reviewer: We have amended Fig 2 as noted above, indicating that Kim et al. was removed from final exclusion after being included in the risk of bias assessment. 

I find the use of questions in the discussion unusual (line 271-272, line 309-310). Perhaps consider these as subheadings for a section if you are unable to revise in another manner?

Response to the Reviewer: We have created subheadings out of our previous queries. This divides the discussion more effectively, and we thank the reviewers for the suggestions in this matter. 

Do the authors view any limitations of their methodological approach? Limitations at this time are limited to the absence of RCTs that met full inclusion criteria and thus an inability to execute the full protocol.

Response to the Reviewer: The absence of moderate to high-quality evidence in RCTs and SRs illustrates the challenge of progressing outcomes-based research as the quality of research evidence has stagnated for 30 years and cannot be used to answer any research question with any level of certainty. Our SR is unable to contribute any clinical recommendations due to the continued lack of high-quality evidence as it related to this specific question. We have included this in our Limitations and have included two new references to support these claims. 

Reviewer #3: Dear authors:

First of all I must congratulate you on the great research work carried out. I think it is a great topic and a great systematic review with all the requirements for a review. I would like to suggest some areas for improvement:

- You should include a flow chart so that we readers can see how you have selected the articles and how many losses you have obtained in this search. I strongly suggest that you insert a specific flowchart for reviews and meta-analyses.

Response to the Reviewer: We have included Figs 1 & 2 as a PRISMA flowchart which demonstrate the overall screening process used for the RCTs examined as part of this review. We did not include SRs, so this is omitted from the flowchart. 

- You talk about conducting a systematic review and meta-analysis study, but you have not conducted the meta-analysis, so in the result section you should include all the data derived from the meta-analysis, as well as the diagrams, graphs and/or figures derived from it. So far, the article would only be a systematic review, so you have two options: leave it as it is and call it a review or add the entire meta-analysis.

Response to the Reviewer: None of the RCTs met the criteria established by our previously published protocol. Our prospective intent was to do a meta-analysis. Our results indicate that no RCTs met the criteria, and our discussion outlines our inability to proceed with the meta-analysis.

---

## [Decision Letter · Decision Letter 1]

2 Jan 2024

Is there ‘trustworthy’ evidence for using manual therapy to treat patients with shoulder dysfunction?: A systematic review

PONE-D-23-32393R1

Dear Dr. Clewley,

We’re pleased to inform you that your manuscript has been judged scientifically suitable for publication and will be formally accepted for publication once it meets all outstanding technical requirements.

Kind regards,

Žiga Kozinc

Academic Editor

PLOS ONE

Additional Editor Comments (optional):

Reviewers' comments:

Reviewer's Responses to Questions

**Comments to the Author**

1. If the authors have adequately addressed your comments raised in a previous round of review and you feel that this manuscript is now acceptable for publication, you may indicate that here to bypass the “Comments to the Author” section, enter your conflict of interest statement in the “Confidential to Editor” section, and submit your "Accept" recommendation.

Reviewer #1: All comments have been addressed

Reviewer #2: All comments have been addressed

2. Is the manuscript technically sound, and do the data support the conclusions?

Reviewer #1: Yes

Reviewer #2: Yes

3. Has the statistical analysis been performed appropriately and rigorously? 

Reviewer #1: Yes

Reviewer #2: N/A

4. Have the authors made all data underlying the findings in their manuscript fully available?

Reviewer #1: Yes

Reviewer #2: Yes

5. Is the manuscript presented in an intelligible fashion and written in standard English?

Reviewer #1: Yes

Reviewer #2: Yes

6. Review Comments to the Author

Reviewer #1: Congratulations for this research. This is a very interesting article.I think that this manuscript is ready for publication.

Reviewer #2: I applaud the author’s efforts to address reviewer comments with this revised submission. Changes to the manuscript have improved clarity and readability. Figure 1 and 2 are logical and convey the steps of the screening process effectively. The subtitle changes guide the reader efficiently through the methods and discussion sections.

7. PLOS authors have the option to publish the peer review history of their article (what does this mean?). If published, this will include your full peer review and any attached files.

Reviewer #1: **Yes: **Paraskevopoulos Eleftherios

Reviewer #2: No

---

## [Editor Report · Acceptance letter]

8 Jan 2024

PONE-D-23-32393R1 

PLOS ONE

Dear Dr. Clewley, 

I'm pleased to inform you that your manuscript has been deemed suitable for publication in PLOS ONE. Congratulations! Your manuscript is now being handed over to our production team.

Kind regards, 

on behalf of

Dr. Žiga Kozinc 

Academic Editor

PLOS ONE